

# An optimized treatment for algorithmic differentiation of an important glaciological fixed-point problem.

Daniel Goldberg[1], Sri Hari Krishna Narayanan[2], Laurent Hascoet[3], and Jean Utke[4]

[1]Univ. of Edinburgh, School of GeoSciences, Edinburgh, United Kingdom
[2]Maths. and Comp. Science Division, Argonne National Lab, Argonne, IL, USA
[3]INRIA, Sophia-Antipolis, France
[4]Allstate Insurance Company, Northbrook, IL, USA
*Correspondence to:* D N Goldberg (dan.goldberg@ed.ac.uk)

**Abstract.** We apply an optimized method to the adjoint generation of a time-evolving land ice model through algorithmic differentiation (AD). The optimization involves a special treatment of the fixed-point iteration required to solve the nonlinear stress balance, which differs from a straightforward application of AD software, and leads to smaller memory requirements and in some cases shorter

computation times of the adjoint. The optimization is done via implementation of the algorithm of Christianson [1994] for reverse accumulation of fixed-point problems, with the AD tool OpenAD. For test problems, the optimized adjoint is shown to have far lower memory requirements, potentially enabling larger problem sizes on memory-limited machines. In the case of the land ice model, implementation of the algorithm allows further optimization by having the adjoint model solve a

sequence of linear systems with identical (as opposed to varying) matrices, greatly improving performance. The methods introduced here will be of value to other efforts applying AD tools to ice models, particularly ones which solve a "hybrid" shallow ice / shallow shelf approximation to the Stokes equations.

## 1   Introduction

In recent decades it has become clear how little we understand about the processes governing ice sheet behavior (*Vaughan and Arthern*, 2007), and the complexity that is required in numerical ice sheet models in order to understand this behavior (*Little et al.*, 2007; *Lipscomb et al.*, 2009). The representation of poorly-understood processes in ice sheet models leads to large, poorly-constrained parameter sets, the size of which might potentially scale with the size of the numerical grid. It is

vital that there be a means to relate the outputs of an ice sheet model back to these parameters, both comprehensively and efficiently. However, the simplest method of sensitivity assessment – running the model multiple times while varying each parameter in isolation – quickly becomes intractable because of the complexity of the models. Consider, for instance, a dynamic model of the Antarctic



Ice Sheet, which takes several days to run on a supercomputing cluster, and contains several hundred

thousand parameters pertaining to the spatially varying frictional and geothermal properties of the bed over which it slides. Assessing the sensitivity of the model to this parameter field by the method described above would not be feasible.

*Adjoint models* provide a means to assess these sensitivities in a way which is independent of the number of parameters. The adjoint of an ice sheet model simultaneously calculates the derivatives of

a single model output (often called a *cost function*) with respect to all model parameters – or rather, the *gradient* of the cost function with respect to the parameter set, or *control variables*. Note that the latter computation more naturally lends itself to scientific inquiry, as

– this single output can be one of societal interest, for instance the contribution of an ice sheet to sea level over a given time window; and

– an investigator is unlikely to solely be interested in just one of these (potentially) several hundred thousand poorly-constrained parameters.

The adjoint model is essentially the linearization of the model, only the information is propagated backward in time (or rather in reverse to computational order). As such, the original model is often referred to as the *forward model*. Essentially, it is this backward-in-time propagation that allows for

simultaneous calculation of these derivatives, regardless of the dimension of the parameter set.

One of the earliest instances of the use of the adjoint of an ice sheet flow model was that of *MacAyeal* (1992), in which a control method was developed to optimally fit a model to observed velocities through adjustment of bed friction parameters. The ice flow model used in this study was a depth-integrated approximation to the shear-thinning Stokes equations, appropriate to ice shelves

and weak-bedded streams (*MacAyeal*, 1989). Moreover, it was a "static" model, i.e. it consisted only of the nonlinear stress balance governing ice velocities, and did not evolve the ice geometry or temperature. The method has since been used in a number of applications (e.g., *MacAyeal et al.*, 1995; *Rommelaere*, 1997; *Vieli and Payne*, 2003; *Larour et al.*, 2005; *Khazendar et al.*, 2007; *Sergienko et al.*, 2008; *Joughin et al.*, 2009). Similar methods have been applied to "higher-order" approximations

(*Pattyn et al.*, 2008), or to the Stokes equations themselves (e.g., *Morlighem et al.*, 2010; *Goldberg and Sergienko*, 2011; *Petra et al.*, 2012; *Perego et al.*, 2014; *Isaac et al.*, 2015).

More recently, algorithmic differentiation (AD) tools have been applied to ice sheet models for adjoint model generation. AD tools differentiate models by differentiating elemental operations and applying the chain rule. They have been applied extensively to atmospheric and ocean codes (*Errico*,

1997; *Heimbach et al.*, 2002; *Heimbach*, 2008). The use of AD offers ease of differentiation of the model. For instance, the majority of the adjoint models mentioned in the previous paragraph ignore the dependence of nonlinear ice viscosity on strain rates, producing an "approximate" set of adjoint equations which have the same form as the forward model, allowing for code reuse. At the same time, this "approximate" adjoint ignores terms in the model gradient without knowing whether they are



negligible. While the "full" adjoint model involves equations distinct from the forward model, the use
of AD avoids having to write the code to solve them. Another advantage is modularity. Modifying,
for example, the specific form of strain-rate dependence of viscosity in an ice sheet model would
then require invasive changes to an analytically-derived set of adjoint equations. When generating
the adjoint through AD, these changes are automatic. Furthermore, AD tools are invaluable when

dealing with time-dependent or multiphysics models, where model complexity makes it very difficult
to generate adjoint code "by hand". In fact, to date the only time-dependent ice sheet adjoint models
have been generated through the use of AD (*Heimbach and Bugnion*, 2009; *McGovern et al.*, 2013;
*Goldberg and Heimbach*, 2013; *Larour et al.*, 2014).

For clarity we will draw a distinction between the partial differential equations (PDEs) that com-

prise a mathematical model of a physical system, and the computational model that discretizes these
equations. The PDEs represent an operator, the linearization of which has an adjoint (the *continuous*
adjoint), which can be discretized *Goldberg and Sergienko* (2011). Alternatively, the computational
model can be differentiated directly. We focus on this *discrete* adjoint in this paper. As mentioned
above, a discrete adjoint model can be thought of as the reverse order computation of the original

model *Griewank and Walther* (2008); *Heimbach and Bugnion* (2009), but an important subtlety is
that this discrete adjoint may not necessarily correspond to the correct continuous adjoint, a subtlety
which bears on the accuracy of ice sheet adjoint models.

Most ice flow models solve a nonlinear elliptic system of partial differential equations (PDEs) for
ice velocity, and these equations require an iterative fixed-point approach. (Here "most ice flow mod-

els" is taken to mean *all* ice flow models, except those which make the Shallow Ice Approximation
(SIA, *Hutter* (1983)). The SIA strictly applies only to slow-moving ice frozen at its base, and not the
fast-flowing ice streams at the Antarctic and Greenland margin which currently exhibit variability.)
We refer to this fixed-point iteration as the Forward Fixed Point Iteration (**FFPI**). Ice sheet models of
this type, to which AD tools have been applied previously, simply step backward through the FFPI

(*Goldberg and Heimbach*, 2013; *Larour et al.*, 2014; *Martin and Monnier*, 2014). This strategy is
sometimes referred to as the *mechanical adjoint* (*Griewank and Walther*, 2008). The mechanical ad-
joint of a fixed-point solution is in fact the iterative solution of a distinct fixed-point problem, whose
convergence differs from that of the forward loop (*Christianson*, 1994), and to which we refer as the
Adjoint Fixed Point Iteration (**AFPI**). As such the mechanical adjoint could potentially perform too

many iterations, thereby wasting resources; or too few iterations, resulting in decreased accuracy. In
fact, in some cases the mechanical adjoint can be inaccurate regardless, as we show in Section 4.1.
Additionally, the mechanical adjoint can lead to burdensome memory and/or recomputation loads as
discussed in Section 3. *Martin and Monnier* (2014) show accuracy can be maintained by truncating
the iteration in the mechanical adjoint, but do not provide a robust, situation-independent way of

doing so.



*Christianson* (1994) provides a mathematical strategy for finding the adjoint of a fixed-point problem via direct solution of a related fixed-point problem. The convergence of this related problem can be directly evaluated, avoiding the problem of too many or two few iterations. A novelty of the approach is that only information from the converged state of the forward loop is used for the adjoint computation, permitting additional efficiency gains. In this paper we present an application of the AD software OpenAD (*Utke et al.*, 2008) to the MITgcm time-dependent glacial flow model (*Goldberg and Heimbach*, 2013). A different AD tool has previously been applied to this ice model, so here we focus on the implementation of the Christianson algorithm (henceforth called **BC94**) – an innovation which is observed to yield substantial improvements in performance.

## 2 Fixed-point problem

The forward model to which AD methods are applied is that of *Goldberg* (2011), which is a "hybrid" of two low-order approximations to the nonlinear Stokes flow equations that govern ice creep over timescales longer than a day (*Greve and Blatter*, 2009). These are the Shallow Ice Approximation, appropriate for slow-flowing ice governed by vertical shear deformation, and the Shallow Shelf Approximation (SSA; *Morland* (1987); *MacAyeal* (1989)), appropriate for fast-flowing ice governed by horizontal stretching and shear deformation. The hybrid equations have been shown appropriate in both regimes, and represent considerable computational savings over the Blatter-Pattyn equations (*Blatter*, 1995; *Pattyn*, 2003; *Greve and Blatter*, 2009), as they require the solution of a two-dimensional system of elliptic PDEs rather than a three-dimensional one.

We do not discuss the details of the model here, as they are given in detail in *Goldberg* (2011) and in *Goldberg and Heimbach* (2013). Rather, we focus on its FFPI. Conceptually, the model algorithm can be divided into two components: prognostic (time-dependent) and diagnostic (time-independent). In the MITgcm land ice model, the prognostic component comprises an update to ice vertical thickness ($H$) through a depth-integrated continuity equation, as well as an update of the surface elevation and, implicitly, the portion of the model domain where ice is floating in the ocean rather than in contact with its bed. The diagnostic component solves the FFPI for ice velocities based on the current thickness profile. Mathematically this step can be understood as the inversion of a nonlinear operator $F$:

$$F(\boldsymbol{u}, \boldsymbol{a}) = \boldsymbol{f}. \tag{1}$$

Here $\boldsymbol{u}$ is a vector representing horizontal depth-averaged velocities $u$ and $v$. $F$ is the discretization of a nonlinear elliptic PDE in depth-averaged velocity. $\boldsymbol{a}$ represents the set of material parameters that determine the coefficients of the PDE: ice thickness ($H$), basal friction rheological parameters ($C$), and ice rheological parameters ($A$). $\boldsymbol{f}$ is the discretization of driving stress (*Cuffey and Paterson*, 2010), or the depth-integrated hydrostatic pressure gradient (which is determined by ice thickness). In this model (and in many others) the nonlinear elliptic equation is solved





by a sequence of solutions of linear elliptic operators, where the operators depend on the result of the previous linear solve:

$$\boldsymbol{u}_{(m+1)} = (L\{\boldsymbol{u}_{(m)}, \boldsymbol{a}\})^{-1}\boldsymbol{f} \equiv \Phi(\boldsymbol{u}_{(m)}, \hat{\boldsymbol{a}}), \tag{2}$$

where, in the definition of $\Phi$, $\hat{\boldsymbol{a}}$ represents the augmentation of the set $\boldsymbol{a}$ to include $\boldsymbol{f}$. $L$ is a linear operator constructed using $\boldsymbol{u}_{(m)}$, the current iterate of $\boldsymbol{u}$, and the parameters $\hat{\boldsymbol{a}}$. Note that $\hat{\boldsymbol{a}}$ will differ for each time step through the dependence on ice thickness, which is updated by the prognostic component of the model. In general, the ice rheological parameters depend on ice temperature, which is advected and diffused over time. Our ice model does not have a thermomechanical component, but once developed, it will not affect the algorithm we present in this paper.

Eq. (2) is our FFPI mentioned previously. In practice the iteration is truncated when subsequent iterates agree in some predefined sense, but in theory will converge to a unique solution $\boldsymbol{u}_*(\hat{\boldsymbol{a}})$. In the process of computing the adjoint to the ice model, $\frac{\partial \boldsymbol{u}_*}{\partial \hat{\boldsymbol{a}}}$ must be found, either directly or indirectly. The focus of this paper is an efficient, scalable method of computing this object.

## 3 Forward model and "mechanical adjoint"

Here we give a brief overview of how the model and its mechanical adjoint are constructed. For further details the reader should consult *Goldberg and Heimbach* (2013). Table 1 contains a high-level pseudocode version of the ice model time stepping procedure. Superscripts denote time step indices. First the velocity solve (CALC_DRIVING_STRESS and the following loop) finds ice velocities based on current ice thickness and material parameters; then the prognostic component updates thickness (ADVECT_THICKNESS). The function $\Phi$ comprises the construction of the linear system $L$ (including the nonlinear dependence of the matrix coefficients on the previous iterate) and its solution.

Table 2 gives an overview of our implementation of the mechanical adjoint. Here we introduce some notation: for a given computational variable $X$, the *adjoint* to $X$, which formally belongs to the dual tangent space at $X$, is denoted $\delta^*X$ (e.g. *Heimbach and Bugnion*, 2009). The algorithm evolves the adjoint variables (e.g., $\delta^*H$) backward in time. These adjoint variables carry with them the sensitivities of the model output to the corresponding forward variables, and the sensitivities are eventually propagated back to the input parameters. Note that the adjoints of the individual (pseudo-)subroutines are given and correspond to the (pseudo-)subroutines of the forward model, mirroring the way the adjoint is actually constructed. Just like the forward model, the adjoint contains an inner loop – this is a specific implementation of the AFPI, which will be discussed in further detail below. As the computation of $\Phi$ involves the solution of a linear system of equations, the adjoint of $\Phi$ involves the solution of the adjoint of that system. Since the matrix $L\{\boldsymbol{u}_{(m)}, \boldsymbol{a}\}$ is self-adjoint, it is easier to calculate this result analytically than for an AD tool to differentiate the linear solver code



– allowing invocation of external "black box" libraries that cannot be differentiated by the tool. This strategy is used by other applications of AD to ice models (e.g., *Martin and Monnier*, 2014).

An important point to be made is that the inner loop in Table 2 is executed as many times as the corresponding inner loop in the forward model ($lastm^{[n]}$), without any checks of convergence. This could lead to under- or over- convergence, as stated previously. Another important aspect is

170 that at each reverse time step, and, importantly, at each iteration of the FFPI, the state of the forward model is required. In particular, every matrix $L\{\boldsymbol{u}_{(m)}, \boldsymbol{a}\}$ must be stored (or recomputed), along with other intermediate variables within the fixed-point loop. The storage and recovery steps are shown explicitly in tables 1 and 2 – and can lead to burdensome memory loads depending on the number of fixed-point iterations taken at each time step.

The mechanical adjoint of our model was first generated using TAF (Transformation of Algorithms in Fortran; *Giering et al.* (2005)), but has subsequently been generated via OpenAD with little further difficulty.

## 4   Fixed point treatment

*Christianson* (1994) presents an algorithm (BC94) for calculating the adjoint of a fixed-point prob-

180 lem that addresses the shortcomings given above, namely the dependence of the termination of the adjoint loop on that of the forward loop, and the requirement to store variables at each iteration of the adjoint loop. Additionally it provides the opportunity for further optimization when applied to a higher-order ice sheet model, as discussed below.

### 4.1   Mathematical basis

For a rigorous mathematical analysis of BC94 the user is asked to consult the original paper. Here we give a brief overview of its mathematical basis. In terms of Eq. (2), consider the converged state of the fixed point problem:

$$\boldsymbol{u}_* = \Phi(\boldsymbol{u}_*, \hat{\boldsymbol{a}}). \tag{3}$$

Consider a total differential of this equation:

$$\delta\boldsymbol{u}_* = \frac{\partial\Phi}{\partial\boldsymbol{u}}(\boldsymbol{u}_*, \hat{\boldsymbol{a}})\delta\boldsymbol{u}_* + \frac{\partial\Phi}{\partial\hat{\boldsymbol{a}}}(\boldsymbol{u}_*, \hat{\boldsymbol{a}})\delta\hat{\boldsymbol{a}}. \tag{4}$$

Rearranging gives

$$\delta\boldsymbol{u}_* = \left[I - \frac{\partial\Phi}{\partial\boldsymbol{u}}\right]^{-1} \frac{\partial\Phi}{\partial\hat{\boldsymbol{a}}}\delta\hat{\boldsymbol{a}}. \tag{5}$$

If the Euclidean operator norm of the square matrix $\partial\Phi/\partial\boldsymbol{u}$ is less than unity then the above is equivalent to

195 $$\delta\boldsymbol{u}_* = \left(I + \partial\Phi/\partial\boldsymbol{u} + (\partial\Phi/\partial\boldsymbol{u})^2 + (\partial\Phi/\partial\boldsymbol{u})^3 + ...\right) \frac{\partial\Phi}{\partial\hat{\boldsymbol{a}}}\delta\hat{\boldsymbol{a}}. \tag{6}$$





Note that in the above series, $\partial\Phi/\partial\boldsymbol{u}$ is always evaluated at the converged solution $\boldsymbol{u}_*$. The above condition on the norm of $\partial\Phi/\partial\boldsymbol{u}$ will not hold in general – but since this is one of the conditions required for convergence to a fixed point, we can expect that it will be satisfied at $\boldsymbol{u}_*$.

From eq. (6) we obtain the desired *adjoint* operator, approximated by a truncated series of length $n$:

$$\delta^*\hat{\boldsymbol{a}} = \left(\frac{\partial\Phi}{\partial\hat{\boldsymbol{a}}}\right)^T \left[I + \left(\frac{\partial\Phi}{\partial\boldsymbol{u}}\right)^T + \left(\left(\frac{\partial\Phi}{\partial\boldsymbol{u}}\right)^T\right)^2 + ... + \left(\left(\frac{\partial\Phi}{\partial\boldsymbol{u}}\right)^T\right)^n\right] \delta^*\boldsymbol{u}_*. \tag{7}$$

The algorithm of *Christianson* (1994) uses a fixed-point loop in order to calculate (7), the convergence criterion of which determines the truncation length $n$. This loop represents an implementation of the AFPI, distinct from the one implemented by the mechanical adjoint. In order to make this distinction explicit, the operator in eq. (7) can be written

$$\sum_{i=0}^{n} \left(\frac{\partial\Phi}{\partial\hat{\boldsymbol{a}}}\right)^T \prod_{k=n+1-i}^{n} \left(\frac{\partial\Phi}{\partial\boldsymbol{u}}\right)^T, \tag{8}$$

where it is understood that in the $i = 0$ term the product sequence evaluates to the identity. It is straightforward to check that the mechanical adjoint (cf Table 2) effectively computes the operator

$$\sum_{i=0}^{n} \left(\frac{\partial\Phi_{(n-i)}}{\partial\hat{\boldsymbol{a}}}\right)^T \prod_{k=n+1-i}^{n} \left(\frac{\partial\Phi_{(k)}}{\partial\boldsymbol{u}}\right)^T, \tag{9}$$

where $\partial\Phi_{(k)}/\partial\boldsymbol{u}$ and similar terms indicate that the gradient is calculated using the variables that have been stored at forward iteration $k$, rather than at the converged solution. It is apparent that this expression can differ from eq. (7) if some iterates are far from the fixed point, or if the gradient of $\Phi$ is sensitive to $\boldsymbol{u}$. In fact, it has been observed in certain cases that a poor choice of initial iterate can lead to inaccurate adjoint calculation. Furthermore, in the mechanical adjoint, the truncation length depends on the number of forward iterations, which may not be related to the convergence of this series. A scheme which truncates this series based on the size of the truncated terms will be more robust.

### 4.2 Implementation in OpenAD

Tables 3 and 4 give an overview of our implementation of BC94 in the MITgcm ice model using OpenAD. High-level changes to the code were necessary, but the subroutines that comprise the action of the operator $\Phi$ were left unchanged. As shown in 3, rather than calling $\Phi$ directly, the loop implementing the FFPI calls a subroutine called `PHISTAGE` with an argument `phase` which has values `PRELOOP`, `INLOOP`, or `POSTLOOP`. Just before the fixed-point loop `PHISTAGE` is called with `PRELOOP`, which does nothing (that is, nothing in forward mode). Within the loop, `PHISTAGE` is called with argument `INLOOP`, which essentially has the same effect as the call to $\Phi$ in the original ice model time stepping algorithm. After the loop is converged, `PHISTAGE` is called with argument



POSTLOOP, which calls $\Phi$ one more time (which, if the iteration is converged, should have negligible effect). Of key importance is that any storing of variables that takes place within the call to $\Phi$ in the INLOOP phase is *undone* at the end of each iteration, unless convergence is reached. In other

words, exactly one "iteration's worth" of storage occurs during the time step.

The reason for the addition of this layer PHISTAGE is rooted in the nature of OpenAD source transformation. To implement BC94 using this tool, it was found to be simplest to replace OpenAD-transformed code with handwritten code, which can be done at the subroutine level using *templates* files. Such a template was written for PHISTAGE in order to implement the pseudocode in tables

3 and 4. The subroutine thus serves as a "layer" which does not affect the diagnostic ice physics represented by the function $\Phi$ or the prognostic physics implemented outside of the FFPI loop. Thus the modularity offered by the AD approach is not lost.

Table 4 shows how the adjoint model is constructed, making use of the OpenAD-generated adjoint code for $\Phi$. In adjoint mode, the calls to PHISTAGE happen in reverse order. The variable

$\boldsymbol{w}$ is a placeholder with no real role in the forward computation, but the adjoint of the call to PHISTAGE in the POSTLOOP phase assigns to $\delta^* \boldsymbol{w}$ the adjoint values of velocity resulting from AD_ADVECT_THICKNESS. In the INLOOP phase $\delta^* \boldsymbol{w}$ is updated according to the equation:

$$\delta^* \boldsymbol{w}_{(m+1)} = \delta^* \boldsymbol{w}_{(m)} \left( \frac{\partial \Phi}{\partial \boldsymbol{u}} \right)^T + \delta^* \boldsymbol{u} \tag{10}$$

where $m$ indicates the AFPI iteration step. (In the table, the subscript indices are left off for clarity.)

This loop iteratively constructs the truncated infinite series in eq. 7 (or rather, its action on $\delta^* \boldsymbol{u}_*$). Finally, the adjoint-mode call to PHISTAGE with PRELOOP represents the operation of $\left( \frac{\partial \Phi}{\partial \hat{\boldsymbol{a}}} \right)^T$ on the result.

The introduction of the variable $\boldsymbol{w}$ represents the bulk of the modifications that were necessary to implement the algorithm using OpenAD. The only additional modification is a handwritten evalua-

tion of convergence of $\delta^* \boldsymbol{w}$: we terminate when the relative reduction in the *sup*-norm of the change in $\delta^* \boldsymbol{w}$ is below a fixed tolerance. We emphasize that all of these modifications are at the level of the "wrapper" PHISTAGE, which does not contain any representation of model physics (and hence changes to model physics would not impact this subroutine nor its handwritten adjoint code).

### 4.3   Optimization of linear solve

As mentioned previously, evaluating $\Phi$ involves the solution of a large (self-adjoint) linear system, and thus the adjoint of $\Phi$ involves the solution of a linear system with the same matrix (assuming the same values of $\boldsymbol{u}$ and $\hat{\boldsymbol{a}}$). In the mechanical adjoint model, within a given time step, this matrix differs with each iteration of the adjoint loop; however, in BC94, only the right hand side differs. This invariance suggests the use of a linear solver whose cost can be amortized over a number of

solves, such as an L-U decomposition or an algebraic multigrid preconditioner, the internal data structures of which only need be constructed once. In this study, we consider only an L-U solver.





## 5  Test Experiment

A simple experimental setup was developed to test the accuracy, performance, and convergence properties of the implementation of BC94. The setup consists of an advancing ice stream and shelf in a rectangular domain $(x, y) \in [0, 80\mathrm{km}] \times [0, 40\mathrm{km}]$. We prescribe an idealized bedrock topography $R$ and initial thickness $h_0$. $R$ does not vary in the along-flow $(x-)$ direction and forms a channel through which the ice flows, prescribed by

$$R(x, y) = -600 - 300 \times \sin\left(\frac{\pi y}{40\mathrm{km}}\right), \tag{11}$$

while initial thickness is given by

$$h_0(x, y) = \begin{cases} 300\ \mathrm{m} + \min\left(1, \left(\frac{x - 50\ \mathrm{km}}{62\ \mathrm{km}}\right)^2\right) \times 1000\ \mathrm{m} & 0 \leq x < 50\ \mathrm{km} \\ 300\ \mathrm{m} & 50\ \mathrm{km} \leq x \leq 70\ \mathrm{km}. \end{cases} \tag{12}$$

Where $x > 70$ km, there is open ocean (until the ice shelf front advances past this point). Where ice is grounded, a linear sliding governs basal stress:

$$\boldsymbol{\tau}_b = -C\boldsymbol{u} \tag{13}$$

where $C = 25$ Pa $(\mathrm{a}^{-1}\mathrm{m})$. The Glen's Law coefficient (which controls the ice stiffness) is given by $8.5 \times 10^{-18}$ Pa$^{-3}$ a$^{-1}$, corresponding to ice with a uniform temperature of $\sim$-34°C. At the upstream boundary, ice flows into the domain at $x = 0$ at a constant volume flux per meter width of $1.5 \times 10^6$ m$^2$/a. At $y = 0$ and $y = 40$ km no-flow conditions are applied. Velocity, thickness and grounding line are plotted in Fig. 1(a). Further details of the equations are given in *Goldberg and Heimbach* (2013).

In the experiment, a cost function $J$ is defined by running the model forward in time for 8 years, and evaluating the summed square velocity at the end of the run. That is,

$$J = \sum_{i,j} u(i,j)^2 + v(i,j)^2 \tag{14}$$

where $i$ and $j$ indicate cell indices in the $x-$ and $y-$directions, respectively, and $u$ and $v$ are cell-centered surface velocities. Unless specified otherwise time step is 0.2 years and grid resolution is 2000 m, so $1 \leq \mathrm{i} \leq 40$ and $1 \leq \mathrm{j} \leq 20$. The control variable consists of basal melt rate $m$, defined for each cell and considered constant over a cell and in time (and nonzero only where ice is floating), and set uniformly to zero in the forward run, even under floating ice. Fig. 1(b) plots the adjoint sensitivities of $m$, or alternatively $\partial J / \partial m_{ij}$, where $m_{ij}$ is melt rate in cell $(i, j)$. The field shows broad-scale patterns that are physically sensible: in the margins of the ice shelf toward its front, thinning through basal melting will weaken the restrictive force on the shelf arising from tangential stresses at the no-slip boundaries. The driving force for flow is proportional to ice shelf thickness, and so in the center of the shelf thinning leads to deceleration. Meanwhile, ice shelf velocities are very insensitive to melting at the center of the ice shelf front.





We find that the results of the mechanical adjoint and of the adjoint model implementing BC94 (which we henceforth refer to as the "fixed-point adjoint") are almost identical, with a relative difference no larger than $10^{-6}$ over the domain (not shown). However, the adjoint sensitivities should also be compared against a "direct" computation of the gradient, i.e. one which does not involve the adjoint model. In this case $\partial J / \partial m_{ij}$ is approximated through finite differencing, by perturbing $m_{ij}$ by a finite amount and running the forward model again. This calculation is carried out for each cell $(i, j)$. Fig. 1(c) plots $disc_{\mathrm{fd}}$, given by

$$disc_{\mathrm{fd}} = \frac{\delta^* m_{ij}^{fp} - \delta^* m_{ij}^{cd}}{\delta^* m_{ij}^{cd}}, \tag{15}$$

where $\delta^* m_{ij}^{cd}$ is a centered-difference approximation:

$$\delta^* m_{ij}^{cd} = \frac{1}{2\epsilon}(J(m_{ij} + \epsilon) - J(m_{ij} - \epsilon)), \tag{16}$$

and $J(m_{ij} + \epsilon)$ indicates that the meltrate *at cell $(i, j)$ only* is perturbed by $\epsilon$. $\epsilon$ is set to 0.01 m/a uniformly.

$disc_{\mathrm{fd}}$ is seen to become quite large, on the order of $\sim 1\%$ in some parts of the domain, warranting further examination. An implicit assumption in the discrepancy measure $disc_{\mathrm{fd}}$ is that the finite difference approximation has negligible error, which may not be the case. We can estimate where this finite-difference error will be large: from a Taylor series expansion, and ignoring round-off error (which we do not attempt to estimate), the error in approximating the adjoint sensitivity of $m_{ij}$ by finite difference is roughly proportional to the second derivative $\partial^2 J / \partial(m_{ij})^2$. As a proxy for this quantity we plot in Fig. 1(d) the 2nd-order difference of $J$:

$$\Delta^2 J_{ij} = J(m_{ij} + \epsilon) + J(m_{ij} - \epsilon) - 2J \tag{17}$$

Aside from the left-hand boundary, this measure appears to correlate well with $disc_{\mathrm{fd}}$. Thus we can at least partly attribute the pattern of discrepancy in Fig. 1(c) to errors in the finite difference approximation. We emphasize that (17) is not an accurate measure of the second derivative – which is obviously not achievable through finite differencing if first-order derivatives are inaccurate – but is simply meant to give an indication of its magnitude.

### 5.1 Truncation errors

The analysis of *Christianson* (1994) suggests the error of the calculated adjoint depends linearly on both the *reverse truncation error* and the *forward truncation error*. The reverse truncation error is the difference between the final and penultimate iterates in the adjoint loop, i.e. the error associated with terminating the loop after a finite number of iterations. That is, referring to Table 4, if $m$ iterations are carried out, the reverse truncation error is equal to

$$\alpha \| w_m - w_{m-1} \|, \tag{18}$$



where $\alpha$ is related to the gradient of $\Phi$ at the fixed point. The norm here is the $sup$-norm, because this is the norm on which our convergence criterion is based.

While a tight bound for $\alpha$ will vary with each time step, it can be expected that the reverse truncation error will vary linearly with the convergence tolerance and we do not address it further. However, we investigate the dependence on forward truncation error as follows. A sequence of adjoint model runs is carried out with increasingly smaller tolerances for the forward fixed-point iteration loop. The tolerance of the reverse loop is kept at a small value ($10^{-8}$). The adjoint sensitivities corresponding to the smallest forward tolerance ($10^{-9}$) are assumed to be "truth"; error is estimated by comparison with these values. Fig. 2 plots the maximum error in the adjoint calculation over the domain against the forward tolerance, which is a good measure of the forward truncation error. Within a range of forward truncation error the dependence is nearly linear, although this dependence appears to become weaker as forward truncation error becomes smaller.

## 5.2 Performance

Here we evaluate the relative performance of the mechanical and fixed-point adjoint models in terms of both timing and memory use. The results are presented in Table 5, but we must first briefly discuss how the OpenAD-generated adjoint computes sensitivities for a time-dependent model. As mentioned in the introduction, adjoint computation takes place in reverse. This presents an issue, because at each time step in this reverse computational mode, the adjoint model requires knowledge of the full model state at the corresponding forward model time step. In general, keeping the entire trajectory (including intermediate variables) of a time-dependent model run in memory is not tractable. Therefore efficient adjoint computation is a balance between recomputation (beginning from intermediate points in the run known as "checkpoints"), storage of checkpoint information on disk, and keeping variables in memory (in data structures called "tapes"). The "store" and "restore" commands in tables 1-4 refer to tape manipulation. For further information on adjoint computation see *Griewank and Walther* (2000) and *Griewank and Walther* (2008).

In our implementation this amounts to an initial forward run with no taping (aside from the final time step), but writing of checkpoints to disk. This initial run is referred to below as the "forward sweep". Afterwards the "reverse sweep" begins, beginning with the final time step. The reverse sweep consists of an intial adjoint computation for the final timestep. As reverse computation proceeds, the model is restarted from checkpoints to recover variables used in adjoint computation. The details of this process are important because they determine how many extra forward time steps (without taping) must be taken. These plain time steps set up the computation of a subsequent time step in "tape mode", i.e. they write intermediate variables to tape during computation. This is followed immediately by a time step computation in "adjoint mode". In the model runs we consider, no extra plain checkpoints are required. A run of 40 time steps, then, will consist of nearly 40 time steps in "plain mode" (no taping, but with checkpoint writing), 40 time steps in tape mode, and 40



time steps in adjoint mode. Even if adjoint time steps and writing to disk and to tape are negligible, such a run will still take about twice as long as the forward model.

In Table 5 we compare run times for the forward and reverse sweeps for the mechanical and fixed-point adjoints of our test problem, at multiple grid resolutions. We also give run times for the "untouched" model, i.e. code which has not been transformed by OpenAD. The difference between this time and the forward sweep represents writing checkpoints to disk, taping in the final time step, and any other extra steps or changes (e.g. modified variable types) caused by the transformation.

We also show the maximum length of the double tape in memory. There are different tapes for different variable types: integer, double, logical and character. The double tape is observed to require the most memory in our tests. However, due to storage of loop indices, the integer tape is nonnegligible, requiring between 20% (in the largest test) to 50% (in the smallest test) of the memory required by the double tape. The numbers reported represent an upper bound, as our system of reporting tape lengths has a granularity of $16 \times (1024)^2$ elements.

In all cases, the forward and adjoint fixed-point tolerance thresholds are set to $10^{-8}$. As resolution increases, stability considerations require smaller time steps, so the number of time steps doubles when cell dimensions are halved. The simulations are run on Intel Xeon 2.67GHz cpus and the number of cores used is displayed. Unless otherwise specified, the Conjugate Gradient solver from the PETSc library (http://www.mcs.anl.gov/petsc) with IL-U preconditioner is used to invert all matrices. The results show that without further optimization, the BC94 algorithm does not offer large timing performance gain over the mechanical adjoint. The forward sweep is slightly shorter, but the reverse sweep is roughly the same. However, the memory load is far less, only going up to (at most) 136 MB in the high resolution run where the mechanical adjoint uses 2.76 GB. This provides a possible explanation for the forward sweep of the mechanical adjoint being slower: it is overhead associated with the additional memory allocation. As even at the highest resolution this is still a modestly-sized problem, it is likely that certain setups of the model on certain machines would quickly reach memory limits and either crash or beginning swapping memory, significantly affecting performance.

Substantial timing performance gains are not seen until the L-U optimization described in Section 4.3. As discussed, this optimization is made possible by the BC94 algorithm. At the highest resolution tested, the reverse sweep takes 40% less time, and overall the model run is 30% shorter. The performance gain is due to the fact that in a time step, the direct L-U decomposition is only done once, and subsequent linear solves are by forward- and back-substitution, which are far less expensive operations. As indirect solvers such as Conjugate Gradients are typically faster than direct matrix solvers, it is unclear what relative performance gain would be at even higher resolutions; but in the three resolutions tested, relative performance improves with resolution.

We mention that the BC94 algorithm has recently been implemented in the AD tool Tapenade, through a different user interface that relies on directives inserted in the code rather than on the





OpenAD templating mechanism. It has not been tested on an ice flow model but on two other CFD
codes, without our linear solver optimisation part. Their performance results are in line with ours,
with a minor run-time benefit but a major reduction of memory consumption (*Taftaf et al.*, 2015).

## 6  Realistic Experiment

In addition to idealized experiments, the fixed-point adjoint has been tested in a more realistic
setting. Smith Glacier in West Antarctica is a fast-flowing ice stream that terminates in a floating
ice shelf. In recent years, high thinning rates of Smith have been observed (*Shepherd et al.*, 2002;
*McMillan et al.*, 2014), and this is thought to be related to, or even caused by, thinning of the ad-
jacent ice shelves by submarine melting (*Shepherd et al.*, 2004). Here we examine this mechanism
using the fixed-point adjoint. To initialize the time-dependent model, we choose a domain and a rep-
resentation of the bedrock elevation and ice thickness in the region from BEDMAP2 (*Fretwell et al.*,
2013) and constrain the hidden parameters of the model (basal frictional coefficient field and depth-
averaged ice temperature) according to observed velocity using methods that have become standard
in glaciological data assimilation (e.g., *Joughin et al.*, 2009; *Favier et al.*, 2014). The observed ve-
locities come from a dataset of satellite-derived velocity over all of Antarctica (*Rignot et al.*, 2011).

Using the bed and thickness data, and the inferred sliding and temperature fields, the model is
stepped forward for 5 years with 0.2 year time steps. The simulation is run on 24 cpus. As with our
test experiment, submarine melt rate is used as the control variable. The cost function, rather than
being a measure of velocity, is the loss of *Volume Above Floatation* (VAF) in the domain at the end
of the 5 years. VAF is essentially the volume of ice that could contribute to sea level change, and is
often used to assess the effects of ice shelf thinning on grounded ice *Dupont and Alley* (2005). It is
given by


$$\mathrm{VAF} = \sum_i \mathrm{HAF}(i) \Delta x \Delta y, \tag{19}$$

$$\mathrm{HAF}(i) = \left( h(i) + \frac{\rho_w}{\rho} R(i) \right)^+, \tag{20}$$

where $i$ is cell index, $h$ is thickness, $\rho$ and and $\rho_w$ are respectively ice and ocean density, $R$ is bedrock
elevation, and the "+" superscript indicates the positive part of the number. We use $\rho = 918$ kg/m$^3$
and $\rho_w = 1028$ kg/m$^3$. A key aspect is that any floating ice does not contribute to VAF.

The results are shown for the ice shelves connecting to Smith Glacier in Fig. 3, overlain on
grounded ice velocities (adjoint melt rate sensitivities are zero where ice is grounded). It is inter-
esting to note where the sensitivities are largest, along the margins of the ice shelves and also along
the boundary between the two main sections of the ice shelf. The mechanism is similar to that of
our test experiment: the margins are where shear stress is exerted, and thinning here will lessen the



backforce on grounded ice. The boundary between the two sections of the ice shelf likely plays a similar role in the ice shelf force balance, as velocity shear is large in this area (not shown).

Regarding accuracy, the finite-difference approximation to the gradient cannot be found for every ice shelf cell. However, we compared the adjoint sensitivity to the finite difference approximation at
4 arbitrary locations, and relative discrepancy was on the order of $10^{-5}$. In terms of performance, this is a much larger setting than even the highest resolution examined in the test problem. The 500 m cell size leads to approximately 200,000 ice-covered cells in the domain (which means the matrices involved, which incorporate both $x-$ and $y-$ velocities, have 400,000 rows and columns). The forward sweep has a run time of 1700 seconds and the reverse sweep 2340 seconds. (Multiple
runs on the same cluster give similar timing results.) Only the fixed-point adjoint with an L-U solver for the adjoint loop is considered. The timing results are encouraging, indicating that the reverse sweep timing comes closer the forward sweep timing with larger-scale simulations.

## 7   Discussion and conclusions

The fixed-point algorithm of *Christianson* (1994) has been successfully applied to the adjoint cal-
culation of a land ice model. The algorithm is very relevant to the model code, as the bulk of the model's computational cost is the solution of a nonlinear elliptic equation through fixed-point iteration. As many land ice models solve a similar fixed-point problem – particularly those intended to simulate fast-flowing outlet glaciers in Antarctica and Greenland – the methodology introduced here has potential for the application of algorithmic differentiation techniques to other ice models.
The implementation of the algorithm replaces a small portion of AD-generated code by handwritten code. However, this is done such that it does not interfere with the modularity offered by AD approach, and it does not require revision as model physics change.

The algorithm offers two advantages over the more straightforward "mechanical adjoint," i.e. the application of AD without intervention. First, the code solves the true adjoint to the fixed point it-
eration, rather than an approximation (*c.f.* Eq. 9). This avoids inaccurate results arising from "bad" initial guesses, and ensures proper convergence of the fixed-point adjoint. Second, the memory requirements do not increase with the number of adjoint iterations as they do with the mechanical adjoint. In the case of OpenAD, the effect on timing performance is small; but for machines with limited memory or for larger problems, the large memory load associated with the mechanical ad-
joint will be a serious issue.

In the context of our ice model, the nature of the algorithm allows for further optimization, as it replaces the sequential solve of linear systems with differing matrices to a sequence of solves with the same matrix. Replacing the Conjugate Gradient solver of the forward model with a direct L-U solver in the adjoint model leads to further performance improvement. The ratio of the reverse
sweep to forward sweep, which is roughly the ratio of the run times of adjoint and forward models,



decreases from 2.6 for the smallest problem considered to 1.4 for the largest. In the case where only a single time step is taken (not discussed above), no checkpoints are necessary, and the duration of the reverse sweep can be as little as 0.3 times the forward sweep.

It should be pointed out that some authors have implemented ice model adjoint generation with-
out any iteration within the adjoint model. Depending on the approximation to the Stokes momentum balance used, the adjoint stress balance can be derived directly from the equations involved (*Perego et al.*, 2014; *Isaac et al.*, 2015). The result is a linear elliptic equation that can be solved without iteration, but which leads to a linear system that is far less sparse than in the forward model. Additionally, the equations must potentially be re-derived if the model physics are
changed. Moreover, not all such approximations to the Stokes balance allow such an approach. "Hybrid" stress balances, which solve two-dimensional approximations to the Stokes balance and are appropriate for both fast-sliding and slow creeping flow, are increasing in popularity due to low computational cost but reasonable agreement with the First Order approximation [e.g. *Goldberg* (2011); *Schoof and Hindmarsh* (2010); *Cornford et al.* (2013); *Arthern et al.* (2015); *W. Lipscomb,*
*pers. comm*]. Our ice model implements such a hybrid stress balance. Differentiating such a balance at the equation level is possible but very tedious, and leads to very complicated expressions that depend strongly on discretization (*Goldberg and Sergienko*, 2011), both undesirable properties. Thus we argue that our application of the Christianson fixed-point algorithm in our algorithmically differentiated ice model framework represents a contribution to land ice modeling in general.

**8 Code availability**

All code necessary to carry out the experiments is publicly available through the MITgcm, OpenAD and PETSc websites. Please see the supplement to the paper for detailed instructions regarding installation and running of experiments.

**9 Acknowledgements**

This work was made possible in part through a SAGES (Scottish Alliance for Geoscience, Environment and Society) travel grant for early career exchange, NERC grant NE/M003590/1, and by a grant from the U.S. Department of Energy, Office of Science, under contract DE-AC02-06CH11357. Additionally the authors are grateful for valuable input from B. Smith, J. Brown and P. Heimbach.





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





**Table 1.** Pseudocode version of forward model time-stepping procedure.

```
FOR n = initialTimeStep TO finalTimeStep
    // Constructs â from H^[n] :
    CALL CALC_DRIVING_STRESS(H^[n])
    m = 0
    REPEAT UNTIL CONVERGENCE OF u
        u = Φ(u,â)
        m = m+1
        store L, u and other variables
    lastm^[n] = m
    // Finds H^[n+1] from continuity equation with u:
    CALL ADVECT_THICKNESS()
```

$$u = \Phi(u,\hat{a})$$

$lastm^{[n]} = m$

**Table 2.** Pseudocode version of mechanical adjoint.

```
FOR n = finalTimeStep DOWNTO initialTimeStep
    // Constructs δ*H^[n] and δ*u^[n] from δ*H^[n+1]
    // via the adjoint of the continuity equation :
    CALL AD_ADVECT_THICKNESS()
    REPEAT lastm^[n] TIMES
        restore L, u and other variables
        δ*â = δ*â + δ*u (∂Φ/∂â)^T
        δ*u = δ*u (∂Φ/∂u)^T
    // Updates δ*H^[n] from δ*â :
    CALL AD_CALC_DRIVING_STRESS(δ*H^[n])
```

$$\delta^*\hat{a} = \delta^*\hat{a} + \delta^*u \left(\frac{\partial \Phi}{\partial \hat{a}}\right)^T$$

$$\delta^*u = \delta^*u \left(\frac{\partial \Phi}{\partial u}\right)^T$$





**Table 3.** Pseudocode version of modified forward model for BC94.

```
        FOR n = initialTimeStep TO finalTimeStep

            // Constructs â from H[n] :
            CALL CALC_DRIVING_STRESS(H[n])
            u = initial guess
            CALL PHISTAGE(PRELOOP, w, u, â)
            REPEAT UNTIL CONVERGENCE OF u
                CALL PHISTAGE(INLOOP, w, u, â)
            CALL PHISTAGE(POSTLOOP, w, u, â)
            // Finds H[n+1] from continuity equation with u:
            CALL ADVECT_THICKNESS()

    SUBROUTINE PHISTAGE(phase, w, u, â)

        IF (phase==PRELOOP)
            // do nothing

        ELSE IF (phase==INLOOP)
            save tape pointer
            u = Φ(u,â)
            // Makes sure no storage is done :
            restore tape pointer

        ELSE IF (phase==POSTLOOP)
            u = Φ(u,â)
            store L, u and other variables
```



**Table 4.** Pseudocode version of fixed-point (BC94) adjoint.

```
FOR n = finalTimeStep DOWNTO initialTimeStep

    // Constructs δ*H[n] and δ*u from δ*H[n+1]
    // via the adjoint of the continuity equation :
    CALL AD_ADVECT_THICKNESS()
    CALL AD_PHISTAGE(POSTLOOP, δ*w, δ*u, δ*â)
    REPEAT UNTIL CONVERGENCE OF δ*w
        CALL AD_PHISTAGE(INLOOP, δ*w, δ*u, δ*â)
    CALL AD_PHISTAGE(PRELOOP, δ*w, δ*u, δ*â)
    δ*u = 0.0
    // Updates δ*H[n] from δ*â :
    CALL AD_CALC_DRIVING_STRESS(δ*H[n])

SUBROUTINE AD_PHISTAGE(phase, δ*w, δ*u, δ*â)

    IF (phase==POSTLOOP)
        δ*w = δ*u

    ELSE IF (phase==INLOOP)
        save tape pointer
        restore L, u and other variables
        δ*w = δ*w (∂Φ/∂u)^T + δ*u
        // Makes sure converged state is reused :
        restore tape pointer

    ELSE IF (phase==PRELOOP)
        δ*â = δ*w (∂Φ/∂â)^T
```





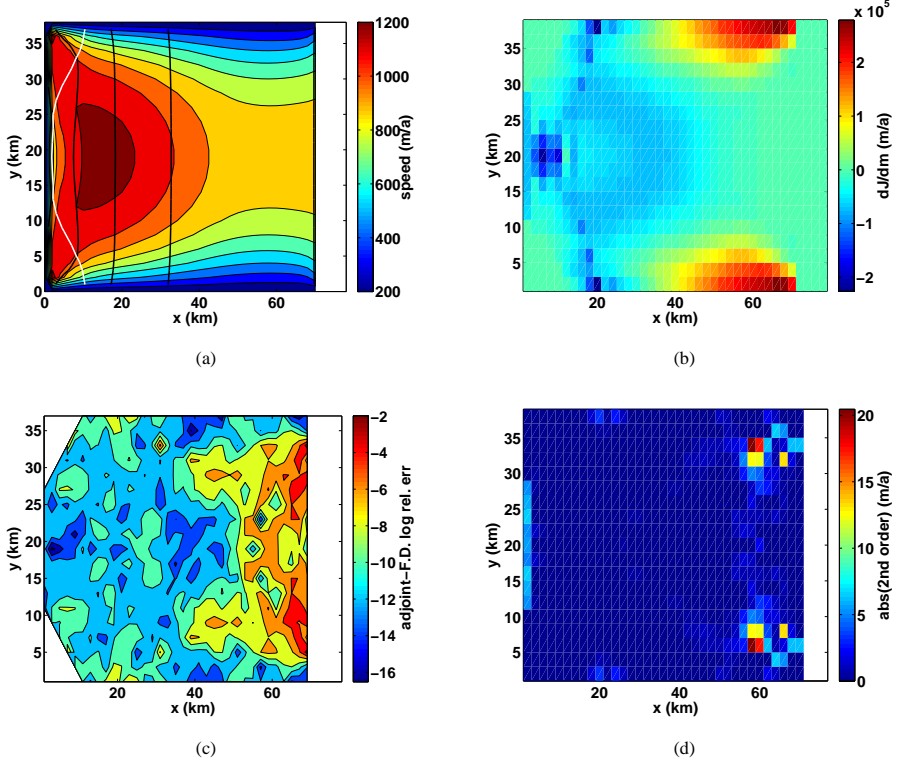

**Figure 1.** (a) Surface speed (shading) in the test experiment. The flow direction is from right to left, and the white portion of the figure is where the ice shelf has not advanced to the end of the domain. Black contours give thickness spaced every 200 m and the white contour is the grounding line. (b) Adjoint sensitivities of ice speed to basal melt rates. (c) (log) relative discrepancy between adjoint sensitivities and the gradient calculated via finite differencing. (d) 2nd order differencing of cost function $J$.





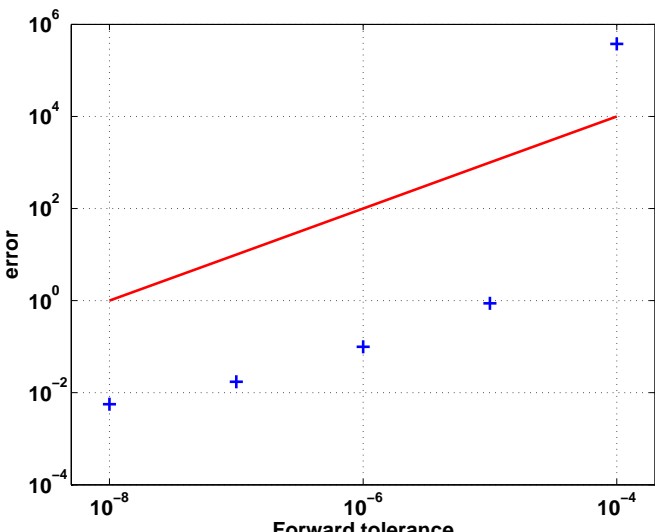

**Figure 2.** Maximum error in fixed-point adjoint calculation versus tolerance of forward loop. The red line indicates linear dependence.

**Table 5.** Timing performance and memory usage of mechanical and fixed-point adjoints. "dbl tape" indicates the length of the double tape.

| grid size | plain (untouched) | | mechanical adjoint | | BC94 algorithm | | BC94 algorithm with L-U optimization | |
|---|---|---|---|---|---|---|---|---|
| **40x20** | **total** | 9.4 s | **total** | 38.9 s | **total** | 37.2 s | **total** | 30.4 s |
| | | | **forward** | 12.2 s | **forward** | 11.7 s | **forward** | 11.7 s |
| **(40 timesteps,** | | | **reverse** | 25.9 s | **reverse** | 25.1 s | **reverse** | 18.3 s |
| **1 cpu)** | | | **dbl tape** | 264MB | **dbl tape** | 8MB | **dbl tape** | 8MB |
| **80x40** | **total** | 110 s | **total** | 434 s | **total** | 425 s | **total** | 321 s |
| | | | **forward** | 134 s | **forward** | 125 s | **forward** | 126 s |
| **(80 timesteps,** | | | **reverse** | 300 s | **reverse** | 300 s | **reverse** | 195 s |
| **1 cpu)** | | | **dbl tape** | 1.38GB | **dbl tape** | 136MB | **dbl tape** | 136MB |
| **160x80** | **total** | 882 s | **total** | 3276 s | **total** | 3204 s | **total** | 2306 s |
| | | | **forward** | 971 s | **forward** | 886 s | **forward** | 886 s |
| **(160 timesteps,** | | | **reverse** | 2297 s | **reverse** | 2304 s | **reverse** | 1417 s |
| **4 cpus)** | | | **dbl tape** | 2.76GB | **dbl tape** | 136MB | **dbl tape** | 136MB |





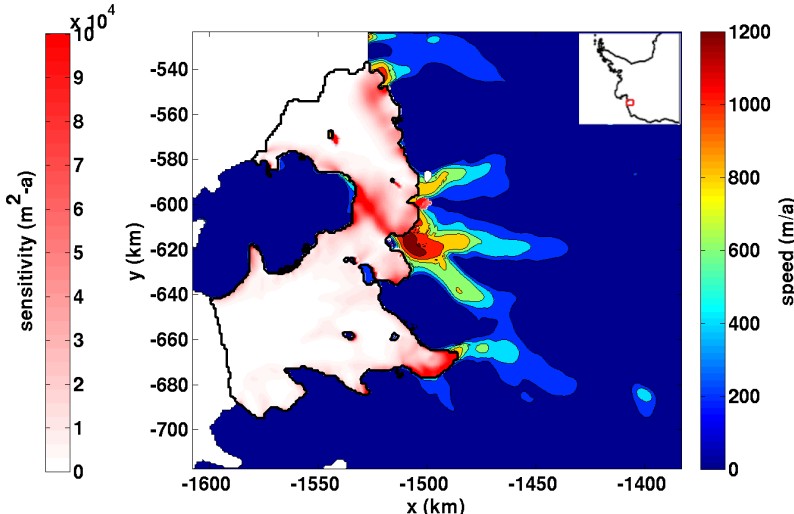

**Figure 3.** Adjoint sensitivity of loss of Volume above Floatation (VAF) to basal melting under the ice shelves adjacent to Smith Glacier (location shown in inset). Filled contours give modeled ice velocity where ice is grounded; red-white shading gives adjoint melt rate sensitivity under ice shelves. The thick black contour denotes the boundary of the ice shelves.

The submitted manuscript has been created by UChicago Argonne, LLC, Operator of Argonne National Laboratory ("Argonne"). Argonne, a U.S. Department of Energy Office of Science laboratory, is operated under Contract No. DE-AC02-06CH11357. The U.S. Government retains for itself, and others acting on its behalf, a paid-up, nonexclusive, irrevocable worldwide license in said article to reproduce, prepare derivative works, distribute copies to the public, and perform publicly and display publicly, by or on behalf of the Government.