# Peer review of "An optimized treatment for algorithmic differentiation of an important glaciological fixed-point problem."

_Geoscientific Model Development, 2016_

## Referee Comment (RC1) · Anonymous Referee #1 · 6 Mar 2016

This paper presents an application of the adjoint fixed point iteration proposed by Christianson (1994) to the MITgcm ice flow model. The paper describes motivates the algorithm, describes the basic principles and the implementation and demonstrates its usability on two problems. Overall, the article is well written and demonstrates the benefits of Christianson's algorithm over applying AD "naively".

I only have a few technical comments:

* Section 5: It would be interesting to state the solver tolerances, and the number of required Picard iterations.

* Lines 463ff: The performance of the LU-solver will degenerate with the size of

[Figure]

the problems, as direct solvers typically scale worse than well-preconditioned linear solvers. Hence, this sentence should be phrased more carefully.

* Figure 2: From the text description it is expected that for a forward tolerance of 10-9, the error would be 0 (as it is assumed as the ground truth). Adding this data point results in a big jump from 10-9 to 10-8. What is the reason for this?

* Table 5: It would be interesting and usefull also list (and discuss) the required Picard iteration numbers that were used in the forward/adjoint solves.

* The references should be checked for correct spelling (e.g. names in titles should be captitalized)

* Lines 469-484: This paragraph is essential and I would have liked to read it earlier. The question that this paragraph adresses is: The adjoint equations are linear, so why does one need to perform a (computationally expensive) Picard iteration at all?

---

## Referee Comment (RC2) · B. Christianson (Referee) · 11 Mar 2016

General comment

I really like this paper.

It's a nice application of an established fixed-point iteration method to a new area, explained in a way that should facilitate use of the approach elsewhere. The iterator used is interesting, because it is sufficiently contractive to converge in a relatively small number of iterations (unlike those used in CFD, for example) but is not super-linear (unlike, say, Newton constructors), and so adjoint iteration is required and the choice of adjoint start-point is potentially significant.

[Figure]

There are two audiences for this paper: geoscientists looking to apply AD efficiently to their specific problems; and those already familiar with AD wanting to apply this particular approach in another application domain. It may be worthwhile to insert additional references to the standard AD literature in order to help the second group get the most out of this paper; and to help the first group learn more about AD, which in turn will give them access to techniques which originated in other application domains.

I make some more specific comments, and suggestions for changes, below: I stress that these really are just suggestions.

Specific comments

line 87: the mechanical adjoint was originally proposed by J.C. Gilbert, "Automatic Differentiation and Iterative Processes", Optimization Methods and Software 1(1) (1992), 13-22, and it might be useful to cite the discussion in Gilbert's paper, as well as that in Christianson 1994. That the mechanical adjoint doesn't always actually solve the adjoint fixed-point problem accurately - or at all - was pointed out by Gilbert: the quick test whether it did is to check if the adjoints corresponding to $u\_0$ are close to zero, where $u\_0$ is the starting value for the forward iterations.

line 155: This would be a good point to insert some references to standard AD literature: as well as the excellent book by Griewank and Walther already in the Reference list, there is a brisk introductory survey paper (available open-access) by Bartholomew-Biggs et al, "Automatic Differentiation of Algorithms", JCAM 124(1-2) (2000), 171-190.

line 164: this observation remains true even when the matrix is not self-adjoint.

line 193: it doesn't have to be the Euclidean norm, contraction with respect to any operator norm will do!

line 202: "uses a fixed point loop to calculate (7)" - not quite. The fixed point loop in Christianson (1994) deliberately calculates (10) rather than (7). This is for two reasons: to avoid repeatedly adding numerically (very) small terms to big ones; and in order to

allow a "warm start" by using an "arbitrary" initial value of delta* w that is close to the fixed point. It may be worth moving equation (10) earlier in the paper and pointing out explicitly that : (a) equation (7) converges with n to the value of delta* a-hat that corresponds to the fixed point of equation (10) ; (b) equation (10) converges to the correct fixed point regardless of what starting point for delta* w is actually used ; and (a) equation (7) corresponds to the result of calculating delta* a-hat after iterating equation (10) precisely n times starting from delta* w = delta* u ; Table 4 seems to assume starting at delta* w = delta* u, but there is no need for this restriction.

line 245: see above discussion on line 202.

line 250: it would be nice to know what norm is being used for the forward convergence: logically the adjoint norm should be used for the reverse convergence. (For example, the sup norm should be used in reverse if the 1-norm is used forwards; the euclidean norm is self-adjoint, etc.) To first order, the error in the calculated value of the cost function J is the inner product of the error in u (from the forward pass) with the converged value of delta* w. This inner product is bounded by any vector norm of the error in u multiplied by the corresponding adjoint vector norm of delta* w: Christianson, "Reverse accumulation and implicit functions", Optimization Methods and Software, 9 (4) (1998), 307-322.

line 393: the point about indirect solvers being more efficient in large dimensions is a good one, but (as well as having the best derivative values) the final forward iteration also generally has the best pre-conditioner.

Technical comments

line 54: applying the chain rule to the numerical values line 76: correspond to a discretization of the correct line 123: of a nonlinear operator F to obtain u: line 165: - this analytic approach allows invocation line 198: required to ensure convergence of Phi to a fixed point line 229: undone at the end of each iteration. Once convergence is reached, storing takes place as normal in the POSTLOOP phase. line 232: simplest

to replace certain specific sections of OpenAD-transformed code line 253: would not require changes to this subroutine [obviously it will affect what the subroutine does!] line 285: i'm really not clear why these are uniformly set to zero line 300: presumably $m_{i,j}^{fp}$ is the value obtained using BC94? line 330: state the range from Figure 2 explicitly here. It would also be useful to have iteration counts for forward and reverse convergence (rather than having to deduce them from Table 5.) line 340: in reverse order relative to forward computation. line 354: recover variable values from the forward computation, so that they can be used in the adjoint computation. line 359: only one level of checkpoints is required. line 442: closer to the forward sweep Fig 1(d) caption: useful to know how the 2nd order differencing was done. Fig 2: seems to have an outlier at 10^-4. Any idea why? Table 5: what is the significance of the red and blue entries?
* * *

---

## Author Comment (AC1) · 22 Apr 2016

We would like to expressly thank Editor Ham, Prof Christianson, and anonymous review for their helpful comments and support of our work. In this document, we address the comments made by the reviewers.

Additionally, it bears mention that due in part to the suggestion that the norm under which adjoint convergence is evaluated should be modified, and due in part to the fact that the computing cluster on which some the experiments in the initial submission were run was replaced by one with differing specifications, all experiments for this paper were re-run. This then accounts for the slightly differing timing estimates in Table 5, where it can be seen that virtually all run times are shorter than their counterparts in the first submission. Also, the time step and # of time steps of the Smith Glacier experienced was slightly modified; this was not for any reason relating to code changes, but simply a wish to have a longer simulation and a shorter time step, which was took less time on the new cluster. Note the shading is different than in the initial submission because (a) the colorscale is different and (b) the duration is longer.

**Reviewer 1**

*This paper presents an application of the adjoint fixed point iteration proposed by Christianson (1994) to the MITgcm ice flow model. The paper describes motivates the algorithm, describes the basic principles and the implementation and demonstrates its usability on two problems. Overall, the article is well written and demonstrates the benefits of Christianson's algorithm over applying AD "naively".*

Thank you very much for your comments!

*\* Section 5: It would be interesting to state the solver tolerances, and the number of required Picard iterations.*

The forward and reverse tolerances are both 10^-8 – stated as well in the original manuscript (line 374). Note that, while this has not changed in the revision, a change has been made to the code in response to referee 2's review (see below) on the importance of using the conjugate norm to establish convergence of the adjoint loop; so for the adjoint this tolerance reflects reduction of the 1-norm and not the sup-norm. Convergence criteria in the forward problem are unchanged.

We now give the average forward and reverse iteration counts as well.

*\* Lines 463ff: The performance of the LU-solver will degenerate with the size of the problems, as direct solvers typically scale worse than well-preconditioned linear solvers. Hence, this sentence should be phrased more carefully.*

Thank you, we make note of this now.

*\* Figure 2: From the text description it is expected that for a forward tolerance of 10-9, the error would be 0 (as it is assumed as the ground truth). Adding this data point results in a big jump from 10-9 to 10-8. What is the reason for this?*

The error presented is absolute error (sup-norm), i.e. not scaled by the norm of the adjoint field. (We now qualify maximum error by maximum *pointwise* error, we feel this is sufficient to avoid confusion). As adjoint values are O(10^5), errors can be considerable even if relative errors are not. Scaling the sup-norm of the error by the sup-norm of the adjoint field would yield a value of ~10^-9 for the leftmost point; and similarly if we considered L2-norms. Still, the linear dependence shown would remain – and it is the linear dependence that we attempt to show with this figure.

Note that the rightmost point (10^-4) is now removed. It was determined that in this experiment, the forward problem was not well-enough converged for the adjoint model to produce a zero-order-correct answer, and we have left this point off in our revision as we feel the four points shown are still sufficient for our argument.

*\* Table 5: It would be interesting and usefull also list (and discuss) the required Picard iteration numbers that were used in the forward/adjoint solves.*

This is done now (see above).

*\* The references should be checked for correct spelling (e.g. names in titles should be captitalized)*

Done, thanks for catching these…

*\* Lines 469-484: This paragraph is essential and I would have liked to read it earlier. The question that this paragraph adresses is: The adjoint equations are linear, so why does one need to perform a (computationally expensive) Picard iteration at all?*

Thank you for the suggestion – this paragraph now appears in the introduction with a brief restatement in the discussion/conclusions section.

**Reviewer 2 (Christianson)**

***General comment***

*I really like this paper.*

*It's a nice application of an established fixed-point iteration method to a new area, explained in a way that should facilitate use of the approach elsewhere. The iterator used is interesting, because it is sufficiently contractive to converge in a relatively small number of iterations (unlike those used in CFD, for example) but is not super-linear (unlike, say, Newton constructors), and so adjoint iteration is required and the choice of adjoint start-point is potentially significant.*

Thank you!

*There are two audiences for this paper: geoscientists looking to apply AD efficiently to their specific problems; and those already familiar with AD wanting to apply this particular approach in another application domain. It may be worthwhile to insert additional references to the standard AD literature in order to help the second group get the most out of this paper; and to help the first group learn more about AD, which in turn will give them access to techniques which originated in other application domains.*

Thank you for the suggestion. The citations you suggest below have been added, and in the introduction as soon as AD is mentioned we give reference to a few general texts (and a community website)

***Specific comments***

*line 87: the mechanical adjoint was originally proposed by J.C. Gilbert, "Automatic Differentiation and Iterative Processes", Optimization Methods and Software 1(1) (1992), 13-22, and it might be useful to cite the discussion in Gilbert's paper, as well as that in Christianson 1994. That the mechanical adjoint doesn't always actually solve the adjoint fixed-point problem accurately - or at all - was pointed out by Gilbert: the quick test whether it did is to check if the adjoints corresponding to $u\_0$ are close to zero, where $u\_0$ is the starting value for the forward iterations.*

Thank you – a reference to the Gilbert paper has been added

*line 155: This would be a good point to insert some references to standard AD literature: as well as the excellent book by Griewank and Walther already in the Reference list, there is a brisk introductory survey paper (available open-access) by BartholomewBiggs et al, "Automatic Differentiation of Algorithms", JCAM 124(1-2) (2000), 171-190.*

Thank you for the suggestion – references added.

*line 164: this observation remains true even when the matrix is not self-adjoint.*

We still believe that a self-adjoint matrix allows for greater ease. To put in context, see Goldberg and Heimbach (2013), Parameter and state estimation with a time-dependent adjoint marine ice sheet model, *The Cryosphere,* 7(6), 1659-1678, eqs 7-8 (the paper is now referenced). Were the matrix not self-adjoint, the subroutine that implements these equations would need to transpose the matrix stored to tape.

*line 193: it doesn't have to be the Euclidean norm, contraction with respect to any operator norm will do!*

We have removed "Euclidean".

Formally, we realise it should be specified that this operator norm is that induced by the norm in which the fixed-point problem converges. On the other hand, since norms in a finite-dimensional vector space are equivalent, then as long as we fix the problem size the choice of norm should not be important. We hope you agree and we do not press the matter further in the text (though see below for line 250).

*line 202: "uses a fixed point loop to calculate (7)" - not quite. The fixed point loop in Christianson (1994) deliberately calculates (10) rather than (7). This is for two reasons: to avoid repeatedly adding numerically (very) small terms to big ones; and in order to allow a "warm start" by using an "arbitrary" initial value of delta\* w that is close to the fixed point. It may be worth moving equation (10) earlier in the paper and pointing out explicitly that : (a) equation (7) converges with n to the value of delta\* a-hat that corresponds to the fixed point of equation (10) ; (b) equation (10) converges to the correct fixed point regardless of what starting point for delta\* w is actually used ; and (a) equation (7) corresponds to the result of calculating delta\* a-hat after iterating equation (10) precisely n times starting from delta\* w = delta\* u ; Table 4 seems to assume starting at delta\* w = delta\* u, but there is no need for this restriction.*

*line 245: see above discussion on line 202.*

Thank you for the correction. While we see the mistake made (eq 7 assumes a specific initial condition) we feel it is correct to say that your algorithm constructs the truncated infinite series within brackets in (7), albeit not as an "end-product", so the wording is amended to reflect this. We feel it is clear enough from (6) that (7) converges to delta\* a-hat, so this is not made explicit.

The discussion around (10) is modified slightly. Attn is brought to the fact this is equivalent to step 9 of Alg 3.1 of BC94, and further we show that the result of n iterations with arbitrary delta\*w0 will converge to the prefactor in (10), showing the point you make that convergence is indep. of the initial guess.

*line 250: it would be nice to know what norm is being used for the forward convergence: logically the adjoint norm should be used for the reverse convergence. (For example, the sup norm should*

*be used in reverse if the 1-norm is used forwards; the euclidean norm is self-adjoint, etc.) To first order, the error in the calculated value of the cost function J is the inner product of the error in u (from the forward pass) with the converged value of delta\* w. This inner product is bounded by any vector norm of the error in u multiplied by the corresponding adjoint vector norm of delta\* w: Christianson, "Reverse accumulation and implicit functions", Optimization Methods and Software, 9 (4) (1998), 307-322.*

Thank you for addressing this. In our first draft (paper and code) the sup-norm was used to evaluate convergence in all loops.

Again, due to the equivalence of norms in finite-dimensional spaces we argue this should not matter too much in practice. Still, we have updated the code to use the conjugate norm (i.e. 1/p+1/q=1 if 1<p<\infty, otherwise sup-norm <--> 1-norm) to evaluate convergence of the adjoint loop. All the results presented in the revision implement this change, with the sup-norm used for the forward iteration. This is made clear in the paper.

(We point out that Fig 2 plots the sup-norm of the error, but results look similar in the 1- and 2-norms.)

*line 393: the point about indirect solvers being more efficient in large dimensions is a good one, but (as well as having the best derivative values) the final forward iteration also generally has the best pre-conditioner.*

This is a good point, though this statement presupposes the type of preconditioner being used – but we do find that as the forward iteration proceeds, the number of required CG iterations for a given accuracy drops. We choose the phrasing:

"Even without the L-U optimization, however, the BC94 algorithm ensures all linear solves in the adjoint model correspond to the converged state of the fixed-point problem. In practice, this matrix is relatively well-conditioned, leading to better performance of the Conjugate Gradient solver."

*Technical comments*

*line 54: applying the chain rule to the numerical values*

done

*line 76: correspond to a discretization of the correct*

done

*line 123: of a nonlinear operator F to obtain u:*

done

*line 165: - this analytic approach allows invocation*

done

*line 198: required to ensure convergence of Phi to a fixed point*

done

*line 229: undone at the end of each iteration. Once convergence is reached, storing takes place as normal in the POSTLOOP phase.*

done

*line 232: simplest to replace certain specific sections of OpenAD-transformed code*

we chose to reword slightly to bring the OpenAD template mechanism to light

*line 253: would not require changes to this subroutine [obviously it will affect what the subroutine does!]*

agreed! But we went with a slightly different modification

*line 285: i'm really not clear why these are uniformly set to zero*

This was simply to more easily define the experiment, as we say now: "(in reality, there would be ``background'' melting to be perturbed, and changes to these melt rates would elicit responses of similar magnitudes, but background melting is zero for the sake of simplicity)"

*line 300: presumably $m_{i,j}^{fp}$ is the value obtained using BC94?*

Yes, noted now

*line 330: state the range from Figure 2 explicitly here.*

done

*It would also be useful to have iteration counts for forward and reverse convergence (rather than having to deduce them from Table 5.)*

Presumably you are referring to table 5, and not the Fig 2 experiments, we agree these values are relevant to table 5 and are now included; however, they are less relevant to figure 2 (and furthermore the experiments would need to be run again to get this information).

*line 340: in reverse order relative to forward computation.*

done

*line 354: recover variable values from the forward computation, so that they can be used in the adjoint computation.*

done

*line 359: only one level of checkpoints is required.*

Done, thank you, it was worded awkwardly before

*line 442: closer to the forward sweep Fig 1(d) caption: useful to know how the 2nd order differencing was done.*

In the caption we refer the reader to eq 17 (now 18).

*Fig 2: seems to have an outlier at 10^-4. Any idea why?*

We determined that the adjoint model essentially was not converging at this high a forward tolerance, and decided to remove this point from the analysis. We feel that the remaining datapoints still support our argument.

***Table 5: what is the significance of the red and blue entries?***

This is to highlight the memory difference between the mechanical and fixed-point adjoint approaches, and to highlight the performance gain of the L-U optimization. We have added a note to the caption.

[revised manuscript text omitted]